# ProEdit: Simple Progression is All You Need for High-Quality 3D Scene Editing

**Jun-Kun Chen    Yu-Xiong Wang**
University of Illinois Urbana-Champaign
{junkun3, yxw}@illinois.edu
immortalco.github.io/ProEdit

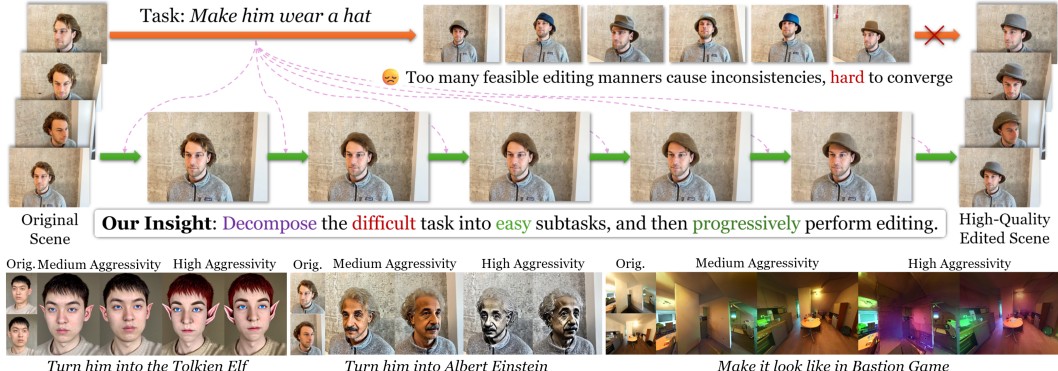

Figure 1: By decomposing a difficult task into easy subtasks and then progressively performing them (upper part), our ProEdit achieves high-quality 3D editing results with bright colors and detailed textures along with introducing new controllability of the editing aggressivity (lower part). **More results are provided on our project page.**

## Abstract

This paper proposes ProEdit – a simple yet effective framework for high-quality 3D scene editing guided by diffusion distillation in a novel *progressive* manner. Inspired by the crucial observation that multi-view inconsistency in scene editing is rooted in the diffusion model's large *feasible output space* (FOS), our framework controls the size of FOS and reduces inconsistency by decomposing the overall editing task into several subtasks, which are then executed progressively on the scene. Within this framework, we design a difficulty-aware subtask decomposition scheduler and an adaptive 3D Gaussian splatting (3DGS) training strategy, ensuring high quality and efficiency in performing each subtask. Extensive evaluation shows that our ProEdit achieves state-of-the-art results in various scenes and challenging editing tasks, *all* through a simple framework *without* any expensive or sophisticated add-ons like distillation losses, components, or training procedures. Notably, ProEdit also provides a new way to control, preview, and select the "aggressivity" of editing operation during the editing process.

## 1   Introduction

The emergence and advancement of modern scene representation models, exemplified by neural radiance fields (NeRFs) [1] and 3D Gaussian splatting (3DGS) [2], have significantly reduced the difficulty associated with high-quality reconstruction and rendering of large-scale scenes. In addition to reconstructing known scenes, there is growing interest in editing existing scenes to create new ones.

38th Conference on Neural Information Processing Systems (NeurIPS 2024).

Among the various editing operations, the *instruction-guided scene editing* (IGSE) stands out as one of the most free-form tasks, supporting editing based on simple text descriptions. Due to the lack of 3D supervision data to train editing models in 3D, current state-of-the-art methods tackle IGSE using *2D diffusion distillation*, which involves distilling editing signals from a pre-trained 2D diffusion model [3, 4]. These methods leverage the 2D diffusion model to edit rendered images of scenes from multiple viewpoints, and then reconstruct the edited scene from these edited images using specific distillation losses.

However, a substantial challenge faced by such distillation-based approaches in achieving high-quality scene editing lies in ensuring that the scene representation converges on the edited multi-view images. Failure to achieve so results in gloomy colors, blurred textures, and noisy geometries (*e.g.*, the failure cases from [5]). We argue that **this challenge is rooted in the diffusion model's large *feasible output space* (FOS) for the same instruction** – since a text instruction can be interpreted in different yet plausible ways. For example, "make the person wear a hat" could be implemented with a hat of any style, shape, size, position, *etc*.

Therefore, large FOS is the underlying cause of *multi-view inconsistency* in 2D editing results, making the scene representation – originally designed for reconstructing from consistent images – hard to converge. Previous work, often unaware of this fundamental issue, deals with multi-view inconsistency by introducing inconsistency-robust distillation losses [6, 7] to tolerant inconsistency, or proposing additional components and training procedures [8, 9] to select consistent images from the FOS. While adding costs and complexities, these methods frequently fail to converge to a high-quality scene when the FOS is considerably large, especially for operations that change the scene's geometry.

In overcoming this challenge posed by the large FOS, *our key insight* is to control the FOS size through *editing task decomposition*, as illustrated in Fig. 1. Building on this insight, we propose *ProEdit*, a simple, novel framework to achieve high-quality IGSE, by decomposing the original, large-FOS task into multiple *subtasks* with significantly smaller FOS, and then *progressively* performing high-quality editing for each of these tasks. With each subtask's FOS effectively controlled, they can be solved under a simple solution *without* the need for additional distillation losses, components, or complex training procedures. Progressively solving all these subtasks naturally leads to a high-quality edited scene that meets the requirements of the original task.

To perform subtask decomposition, we introduce an intuitive formulation of "subtasks" with text encoding interpolation. Based on this formulation, we propose a *subtask scheduler* to determine the subtask decomposition and guide the editing process. This decomposition consists of a sequence of subtasks, where each subtask is applied to the edited scene from the previous one. We adaptively assign subtasks according to the estimated FOS size, so that each subtask has comparable FOS sizes and difficulty levels and can thus be solved relatively easily with high quality and efficiency.

Guided by the subtask scheduler, we progressively iterate on the subtasks to apply editing. Though their FOS size and difficulty are controlled, it still remains non-trivial to make the scene representation converge in precise geometry. Failing to achieve this will accumulate errors across subtasks, leading to unreasonable geometry in the final results. To this end, we choose 3D Gaussian splatting (3DGS) [2] as our scene representation for its high training efficiency. We design a novel *adaptive* Gaussian creation strategy in training to maintain and refine the geometric structure in each subtask, by controlling the size of the splitting and duplication operations. This strategy allows the geometry to be adjusted toward the goal of each subtask, while preventing and removing floc, floating noise, and multi-face structures.

With these key designs, our ProEdit achieves high-quality instruction-guided scene editing in various scenes and editing tasks with precise geometry and detailed textures, as shown in Fig. 1. Notably, ProEdit does not rely on complicated or expensive add-ons, such as specialized distillation losses, additional 3D attention or convolution components, or extended training procedures on the diffusion model. Moreover, as each subtask represents a partial completion of the overall task, our method enables users to *control, preview, and select* the intermediate stages of editing, which we refer to as "aggressivity" of editing operation during the editing process. This can be simply achieved by taking the edited scene from a subtask either during or after the editing process. Thus, in contrast to previous methods such as classifier-free guidance [10] and SDEdit [11], our ProEdit provides a novel way to monitor and manage the editing process. Users can *preview* different versions of editing with the intermediate outcomes, adjust the subtasks *on the fly* accordingly to achieve improved final results, and finally *select* the most satisfactory editing result from all the intermediate ones.

**Our contributions** are three-fold. (1) We offer a novel insight into subtask decomposition and progressive editing, tailored to address the core challenge of large feasible output space in 3D scene editing. (2) We propose a simple yet effective framework, ProEdit, that generates high-quality edited scenes by progressively solving each subtask, without requiring any complicated or expensive add-ons to the diffusion model, while also supporting control, training-time preview, and selection of editing task aggressivity. (3) We consistently achieve high-quality editing results in various scenes and challenging tasks, establishing state-of-the-art performance.

## 2  Related Work

**Learning-Based 3D Scene Representation.**  Our framework necessitates a learnable 3D representation to depict the scene being edited. Traditional methods model the 3D geometric structure of a scene with implicit [12–14] or explicit [15–17] representations. However, these methods require more information or pre-processing beyond multi-view camera images. In 2020, the neural radiance field (NeRF) [1] emerges as the first neural network-based scene representation, enabling direct scene reconstruction from multi-view images captured at known camera locations, inspiring numerous follow-up work [18–24] that explores different aspects including quality, efficiency, and visual effects. Later, 3D Gaussian splatting (3DGS) [2] becomes a new trend, outperforming NeRF and its variants in rendering quality and efficiency. 3DGS also leads to several follow-up variants, aiming to improve geometry [25, 26] and visual effects [27], as well as extending to dynamic 3D scenes [28–30].

**3D Scene Editing.**  Various scene editing tasks have been investigated, each aiming to achieve different editing objectives for a given scene across a range of scene representations. These tasks cover different aspects of a scene, including the location, shape, and color of objects [20, 31–33], physical effects [34], lighting conditions [27, 35, 36], and the overall appearance [5, 7, 9, 37, 38].

**Instruction-Guided Scene Editing.**  Instruction-guided scene editing is a highly free-form yet challenging task, characterized by a straightforward task descriptor – either an editing operation (*e.g.*, "Give the person a hat") or a description of the desired scene (*e.g.*, "A person wearing a hat"). This task has attracted much attention in the computer vision community. Due to the lack of large-scale 3D datasets to train editing models directly in 3D, current state-of-the-art methods [5–7, 9, 38–41] achieve scene editing by distilling knowledge from a pre-trained 2D diffusion model [3, 5] using score distillation sampling (SDS) [42] and its variants. Instruct-NeRF2NeRF (IN2N) [5] and its variants [37, 39] apply SDS-equivalent iterative dataset updates to generate edited multi-view images and train the scene representation on them. One direction of follow-up work [6, 7] proposes novel distillation methods to better utilize the 2D editing capability, while another [9, 41] introduces additional components and training procedures to improve the consistency of generation. However, these approaches are unaware of the core challenge posed by large feasible output space (FOS), mitigating it with add-ons that may still fail when the FOS becomes considerably large. In contrast, our ProEdit is tailored for this challenge by proposing subtask decomposition to explicitly control the size of FOS, thereby extending the capability boundary of instruction-guided scene editing.

## 3  ProEdit: Methodology

The key insight of our ProEdit is to decompose a full editing task, described by a text instruction, into a sequence of simpler subtasks with smaller feasible output space (FOS), and apply each of them progressively on the scene. Our framework consists of three major components: (1) an interpolation-based subtask formulation that defines, obtains, and interprets each subtask; (2) a difficulty-aware subtask decomposition scheduler that breaks down the full editing task into several subtasks of comparable difficulty; and (3) an adaptive 3D Gaussian splatting (3DGS)-based [2] geometry-precise scene editing method that ensures high-quality editing for each subtask, ultimately leading to successful completion of the full task. Our framework is visualized in Fig. 2.

### 3.1  Interpolation-Based Subtask Formulation

In order to decompose a text-described task into subtasks, we first need to clearly define "task" and "subtasks." We define an editing task $T(s, e = E(p))$ as an operation that applies a prompt (instruction) $p$ on the original scene $s$, where $e = E(p)$ denotes the text encoding of $p$ calculated by a frozen text encoder $E(\cdot)$ as part of a 2D diffusion model. The notation $T(s, e)$ represents the edited

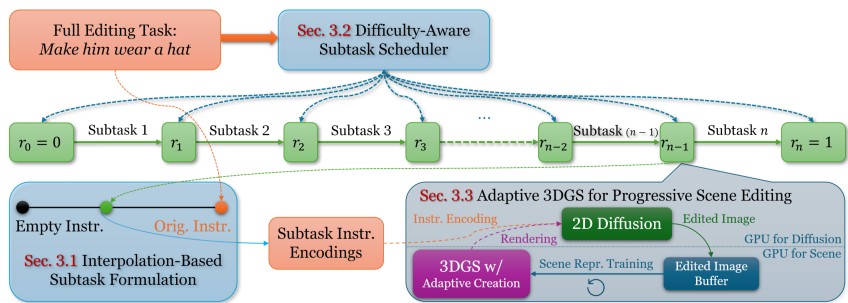

Figure 2: **Our ProEdit framework** features three major designs: an interpolation-based subtask formulation (Sec. 3.1), a difficulty-aware subtask scheduler for subtask decomposition (Sec. 3.2), and an adaptive 3DGS tailored for progressive scene editing through a dual-GPU pipeline (Sec. 3.3). For an editing task, we first decompose it into interpolation-based subtasks to schedule the editing process with the subtask scheduler, and then progressively perform the subtasks with adaptive 3DGS.

scene resulting from this task, and we also use $T(\cdot, e)$ to indicate the mapping from the original scene to the edited scene within this context. Additionally, we define $\varnothing$ as the empty prompt, indicating that the editing task with this prompt retains the original scene, or $T(s, E(\varnothing)) = s$.

Next, we define subtasks as $S(s, r) = T(s, e(r))$ with a ratio $r \in [0, 1]$, where $e(r) = r \cdot E(p) + (1 - r) \cdot E(\varnothing)$. This represents a task characterized by an instruction $p(r) = E^{-1}(r \cdot E(p) + (1-r) \cdot E(\varnothing))$, whose embedding is a ratio-$r$ interpolation between $E(p)$ and $E(\varnothing)$. Assuming that the neural network $E(\cdot)$ is continuous, $S(s, r)$ will also be continuous w.r.t. $r$. Therefore, this formulation provides a continuous space of subtasks or intermediate tasks between the original task $T(\cdot, E(p))$ and the identity mapping $T(\cdot, E(\varnothing))$.

## 3.2 Difficulty-Aware Subtask Scheduler

**Feasible Output Space (FOS) and Task Difficulty.** Inspired by the derivation of SDS [42], we introduce the concept of *feasible output space* (FOS) for an editing task $T(s, E(p))$ as follows: the set of scenes $s'$ such that, when $s'$ is rendered from any view $v$, the resulting image resembles the edited image (based on instruction $p$) from the corresponding view $v$ of the original scene $s$, *i.e.*, the set of all possible scenes that can be regarded as valid edited result for the given task. A larger FOS indicates greater diversity in how the editing task can be executed; however, this variability can cause multi-view inconsistency, if different views are edited differently. Therefore, an editing task with a larger FOS is inherently more difficult to accomplish.

**Formulation of Subtask Decomposition.** Our goal is to decompose the original editing task $T(\cdot, E(p))$ into a sequence of subtasks, such that applying each subtask progressively or iteratively on the current scene leads to the final editing result. Formally, the decomposition of a task $T(\cdot, E(p))$ is a monotonically increasing sequence $r_0, r_1, \cdots, r_n$, where $r_0 = 0, r_n = 1$. We then define $s_i$ as the edited scene resulting from the $i$-th subtask. We have

$$s_i = \begin{cases} s, & \text{(original scene)}, & i = 0, \\ S(s_{i-1}, r_i), & \text{(apply subtask } r_i \text{ on previously edited scene)}, & i = 1, \cdots, n. \end{cases} \quad (1)$$

In other words, the $i$-th subtask is $S(s_{i-1}, r_i)$, which is applied on the edited scene $s_{i-1}$ from the previous $(i-1)$-th subtask. The outcome of the $i$-th subtask is $s_i$.

**Subtask Difficulty Measurement and Approximation.** The difficulty of each subtask $S(s_{i-1}, r_i)$ is measured as being proportional to the size of FOS (a continuous space), which is difficult to compute or even rigorously define. Therefore, we approximate this difficulty by evaluating the difference between the original and edited images of the 2D diffusion model. Intuitively, an editing task that brings a significant change typically has more degrees of freedom, leading to a larger FOS. Additionally, each subtask $r_i$ is applied on the scene $s_{i-1}$, which cannot be determined until all prior subtasks $r_1, r_2, \cdots, r_{i-1}$ are completed. So, we make another approximation based on the assumption that the image of a view in $s_i$ will closely resemble the corresponding view of $s$ edited by the 2D diffusion model following the instruction of the $i$-th subtask. In other words,

$$v_k(s_i) \approx T_{2D}(v_k(s), e(r_i)), \forall k \in V, \quad (2)$$

where $v_k(s)$ is the rendered image at the $k$-th view of scene $s$, and $T_{2D}(v, e)$ is the output of a 2D editing task applied on image $v$ with instruction embedding $e$, generated by the 2D diffusion model. By applying such an approximation to both subtasks and using Learned Perceptual Image Patch Similarity (LPIPS) to measure the perceptual difference between images, we can then define the difficulty metric as

$$\mathrm{d}(r_i, r_j) \overset{\text{Def}}{=\!=\!=} \sum_{k \in V} L_{\mathrm{LPIPS}}(v_k(s_i), v_k(s_j)) \approx \sum_{k \in V} L_{\mathrm{LPIPS}}(T_{2D}(v_k(s), e(r_i)), T_{2D}(v_k(s), e(r_j))). \tag{3}$$

Observing that $\mathrm{d}(r_i, r_j)$'s approximation is only related to the rendered image $v_k(s)$ of the original scene $s$ and is independent of that of the edited scene (namely, $v_k(s_i)$), we can then allow $\mathrm{d}(r_a, r_b)$ to take any two arbitrary subtasks $r_a$ and $r_b$. Our goal is to find the subtask decomposition $r_0, \cdots, r_n$ with similar $\{\mathrm{d}(r_{i-1}, r_i)\}$ for each $i$.

**Difficulty-Aware Adaptive Subtask Decomposition.** The approximation of $\mathrm{d}(r_i, r_j)$ disentangles its computation from the edited scene of task $T(\cdot, e(r_i))$, by substituting it with $T_{2D}(\cdot, e(r_i))$. This enables us to decompose the subtasks from a more *global* perspective. Therefore, we propose an adaptive method to obtain the set of subtask ratios $R = r_0, \cdots, r_n$. The algorithm operates recursively over an interval $[r_a, r_b]$ with a difficulty threshold $\mathrm{d}_{\mathrm{threshold}}$, starting with the interval $[0, 1]$. In each recursion, the algorithm first includes both $r_a$ and $r_b$ in the set $R$, and stops the recursion if $\mathrm{d}(r_a, r_b) \leq \mathrm{d}_{\mathrm{threshold}}$. Otherwise, it selects the middle point $r_m = (r_a + r_b)/2$, and recurses on the intervals $[r_a, r_m]$ and $[r_m, r_b]$. Once the recursion is complete, we obtain the sequence of subtasks $r_0, \cdots, r_n$ by sorting the set $R$, ensuring that $\mathrm{d}(r_{i-1}, r_i) \leq \mathrm{d}_{\mathrm{threshold}}$ for all subtasks.

To simplify the subtask decomposition, we check if there exists a subtask $r_i$ such that $\mathrm{d}(r_{i-1}, r_{i+1}) \leq \mathrm{d}_{\mathrm{threshold}}$. If so, we could safely remove the subtask $r_i$ while still maintaining $\mathrm{d}(r_{i-1}, r_{i+1}) \leq \mathrm{d}_{\mathrm{threshold}}$. This iterative check continues until no further subtasks can be pruned.

Notably, an interpolated subtask can be regarded as a partial completion of the editing instruction. For example, the instruction "Make him smile" with an interpolation ratio of $r = 0.5$ can be interpreted as "Make him half-smile," indicating a lower *aggressivity* of the editing operation. In this context, high aggressivity indicates more significant changes towards the editing operation, whereas low aggressivity reflects greater similarity between the edited scene and the original one. Therefore, our subtask decomposition not only lays the foundation for our editing process but also categorizes task aggressivity, where each subtask corresponds to a specific level of aggressivity. Consequently, beyond performing editing, our ProEdit enables users to control, preview, and select the aggressivity of the editing operation *during or after the editing process*, by utilizing the edited scene of a subtask throughout the progressive editing workflow. Such a capability is absent in previous work.

**Subtask Scheduling.** The subtask scheduler also determines when the current subtask is complete, allowing us to proceed to the next one. Designing an image-based criterion to assess whether the images in the current subtask have been sufficiently edited is challenging. Therefore, we propose a criterion based on the scene representation training procedure. Specifically, when the running mean of the training loss no longer decreases over a specified number of iterations, we regard the scene representation to be converging to the edited scene, indicating that the current editing subtask is complete. Moreover, apart from the subtasks $r_0, r_1, \cdots, r_n$, we prepend an additional subtask $r_0$ to refine the initial scene representation using diffusion-reconstructed original images, and append another subtask $r_n$ to consolidate the editing results, as detailed in Appendix B.

## 3.3 Adaptive 3DGS Tailored for Progression

We choose 3DGS [2] as our scene representation for its high efficiency and rendering quality. However, 3DGS is primarily designed for reconstruction from multi-view consistent images. Directly training on edited images with 3DGS results in a continuously increasing number of Gaussians that overfit the inconsistent views, ending up with an out-of-memory error. Therefore, we propose a novel Adaptive 3DGS specifically tailored for progressive scene editing.

**Basic Workflow for Each Subtask.** As each subtask has a reduced FOS and lower difficulty, we can use a straightforward approach to perform the subtask editing. Consistent with Instruct-NeRF2NeRF (IN2N) [5], we apply a simple iterative dataset update (Iterative DU) that iteratively generates edited views using the diffusion model and employs them to train the scene representation. Unlike NeRFs [1], our 3DGS-based scene representation accepts full images as supervision instead of rays, allowing

us to directly train on the edited images without the need to replace rays. This enables a simpler yet more effective workflow.

**Adaptive Gaussian Creation Strategy.** While the decomposition of subtasks controls the size of FOS and reduces potential inconsistencies, making 3DGS converge on the edited multi-view images remains challenging. Designed only for reconstruction from multi-view consistent images, 3DGS is not robust enough to deal with all inconsistencies. This can lead to overfitting on the inconsistent edited images with view-dependent colors, floating or floc noises, and multi-face structures.

Therefore, we propose an adaptive Gaussian creation strategy to refine the geometry of 3DGS, enabling it to converge on the edited images with reasonable and potentially high-quality geometric structures. As introduced in [2], the original 3DGS maintains Gaussian-represented geometry by periodically culling unnecessary Gaussians based on an opacity threshold, and by creating new Guassians (through splitting or duplicating) to expand model capability according to a training gradient threshold. Our strategy builds on this geometry maintenance schedule by *adaptively* controlling both thresholds. (1) At the beginning of each subtask, we set the opacity of all Gaussians to the threshold and perform several iterations of training without geometry maintenance. This training procedure implicitly identifies the Gaussians that correctly lie on the object surface by making them learn higher opacity, which allows them to be preserved in the scene representation. Conversely, Gaussians with incorrect geometry learn lower opacity and are subsequently culled during the next maintenance phase. (2) To prevent the training process from creating too many noisy Gaussians in a single iteration when operating with edited images, we also control the gradient threshold for Gaussian creation to achieve a smooth increase in the number of Gaussians. We schedule the number of created Gaussians based on the existing Gaussians in the scene and the number previously culled, selecting the threshold according to this scheduled number, as detailed in Appendix D. With these strategies, our 3DGS is able to converge to the edited scenes with clear texture and reasonable, even precise geometry.

**Dual-GPU Training to Decouple Diffusion and 3DGS.** Given the significant difference in iteration speeds – around 2 seconds per generation for the diffusion model inference and less than 0.02 seconds per iteration for the 3DGS training procedure – it is challenging to achieve an effective trade-off on a single GPU during Iterative DU. Inspired by [9, 43], we employ a dual-GPU training schedule to decouple them. The first GPU iteratively generates newly edited images using the diffusion model and stores them in a buffer as the updated dataset. Meanwhile, the second GPU iteratively trains 3DGS with the edited images in the buffer and raises a signal to indicate when the current subtask is complete. This approach enables a highly efficient training procedure within our ProEdit framework.

## 4 Experiments

### 4.1 Experimental Settings

**Scene Representation and Diffusion Model.** As mentioned in Sec. 3.3, our ProEdit leverages 3DGS-based scene representation for high quality and efficiency. We use the Splatfacto model from the NeRFStudio [44] library as our backbone. For the diffusion model, consistent with previous work [5, 6, 37, 45], we use a pre-trained Instruct-Pix2Pix (IP2P) [4] model from HuggingFace.

**Scenes and Editing Instructions.** According to Sec. 3.1, each editing task $T(s, E(p))$ is characterized by a scene $s$ and an instruction $p$, and the desired output is the edited scene. We evaluate our ProEdit on the following scene datasets: (1) The IN2N dataset introduced by Instruct-NeRF2NeRF (IN2N) [5], which is available for free use and is the most widely used dataset in prior work. (2) The ScanNet++ dataset of indoor scenes, released under the ScanNet++ Terms of Use, which is introduced for instruction-guided scene editing in [9]. We use instructions either from previous methods for comparisons or from tasks that require highly noticeable geometric changes in the scene – one of the most challenging editing tasks that previous methods have struggled to perform well.

**Subtask Scheduling.** We determine the number of subtasks to balance editing quality, controllability, and efficiency. For texture-focused instructions (*e.g.*, style transfer), we decompose each task into approximately 4 subtasks using an appropriate threshold $d_{\text{threshold}}$; for geometry-related instructions with much higher FOS, we break each task down into around 8 subtasks with a proper $d_{\text{threshold}}$.

**Baselines.** We compare our ProEdit with recent state-of-the-art instruction-guided scene editing methods, including Instruct-NeRF2NeRF (IN2N) [5] (along with its 3DGS-based implementation

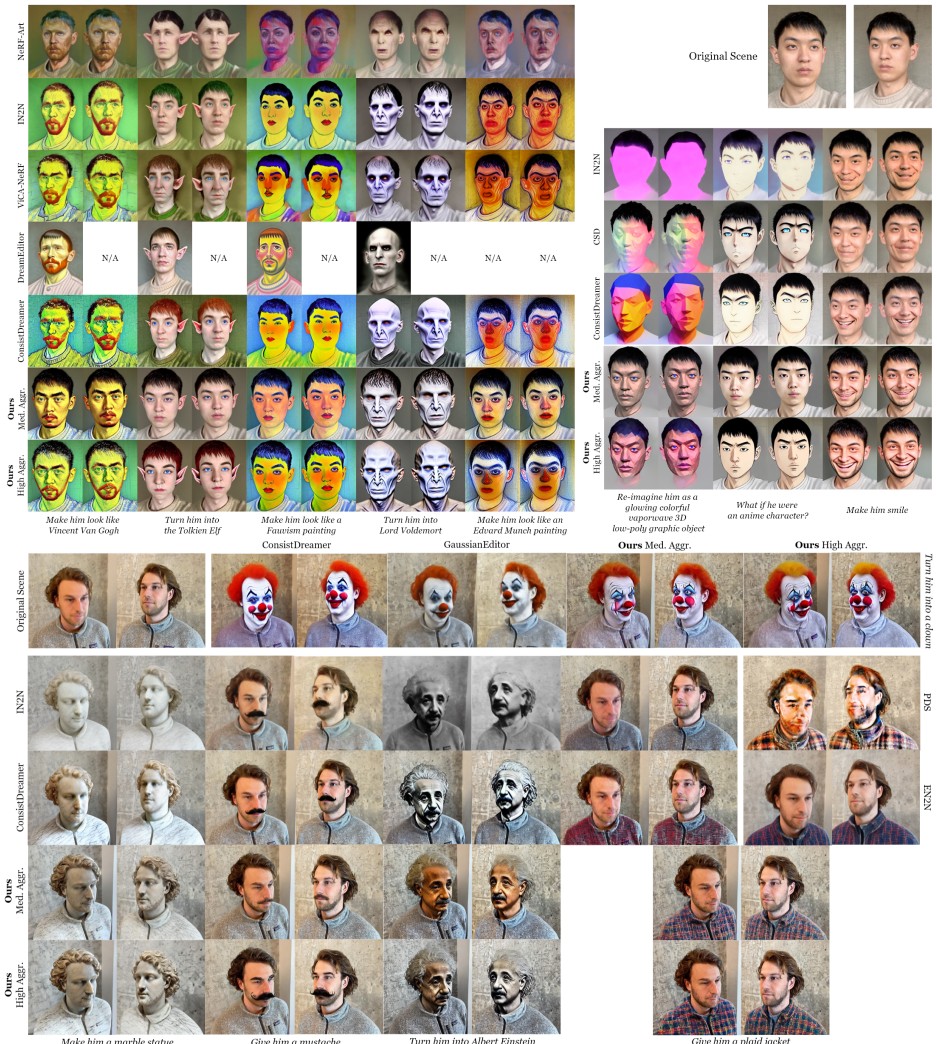

Figure 3: **In the comparative experiments on the Fangzhou and Face scenes**, our ProEdit achieves high-quality editing, with strong instruction fidelity, clear textures, and precise shapes across both levels of aggressivity controlled by subtask scheduling. The "medium aggressivity" editing results are obtained from an intermediate subtask. The editing results of the baselines are sourced from visualizations in their respective papers.

[45]), ViCA-NeRF [41], ConsistDreamer [9], CSD [6], PDS [7], Efficient-NeRF2NeRF (EN2N) [37], DreamEditor [46], *etc*. As different methods use different tasks for visualization in their papers, and some do not provide publicly available code or pre-trained models, our primary comparisons focus on common editing tasks, leveraging the visualizations presented in their papers. Also, we include comparisons for some additional tasks with results generated from available code or re-implementations. As our ProEdit specifically targets the instruction-guided scene editing task, we do not include comparisons with methods designed for other scene editing or generation tasks.

**Implementation Details.** We follow the default hyperparameter settings of the Splatfacto method, and set the classifier-free guidance (CFG) [10] as $7.5 \times 1.5$ for all instructions in the diffusion model. During the editing process for each subtask, consistent with IN2N [5], we use SDEdit's [11] method to control similarity with denoising timesteps between 450 and 850. We also apply HiFA's [47] annealing strategy to gradually decrease denoising timesteps in this process. Utilizing a dual-GPU training workflow (Sec. 3.3), the editing tasks are conducted on two NVIDIA A6000 or A100 GPUs, with each subtask taking 10 to 20 minutes to complete depending on its difficulty and convergence.

**Metrics.** We present the quantitative assessment under the following metrics: User Study of Overall Quality (USO), User Study of 3D Consistency (US3D), GPT Evaluation Score (GPT), CLIP [48]

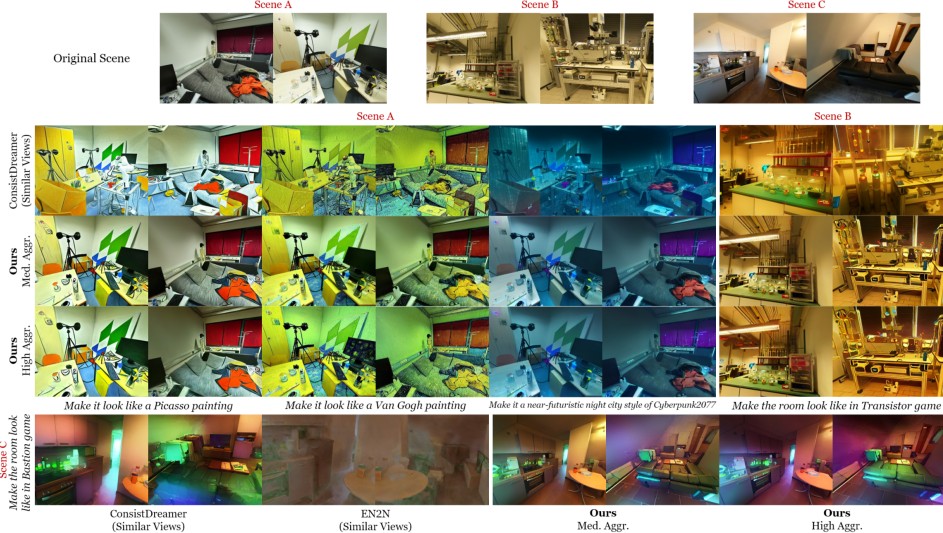

Figure 4: **In the comparative experiments on the ScanNet++ scenes**, our simple ProEdit also achieves high-quality editing that is comparable to, and in some cases even outperforms, the sophisticated baseline ConsistDreamer [9]. All visualizations are sourced from ConsistDreamer's paper.

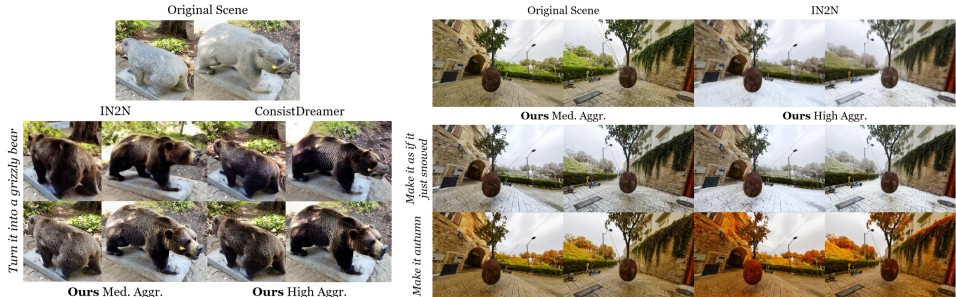

Figure 5: **In the comparative experiments across various outdoor scenes**, our ProEdit not only achieves high-quality editing that surpasses the baselines, but also enables aggressivity controls for a range of scenes and tasks.

Text-Image Direction Similarity (CTIDS), and CLIP Direction Consistency (CDC). The user study was conducted with 26 participants. The GPT score is detailed in Appendix E. The CLIP-based scores are consistent with those reported in IN2N [5].

## 4.2 Experimental Results and Analysis

**Qualitative Results.** Fig. 3 shows the comparisons in the Fangzhou scene and the IN2N's Face scene. Our ProEdit demonstrates results on two levels of editing aggressivity: high aggressivity results are obtained by executing all subtasks, while medium aggressivity results are derived from completing only the first 40% subtasks. Overall, our ProEdit produces high-quality editing results characterized by clear textures, bright colors, reasonable and precise geometry, and high instruction fidelity. Compared with the baselines, our ProEdit shows enhanced geometry editing capabilities, particularly in the "Tolkien Elf" editing which features a thinner face, and the "Lord Voldemort" editing which incorporates more wrinkles in the Fangzhou scene. By contrast, the baselines tend to maintain geometry more similar to the original scene. Notably, for the editing task "Give him a plaid jacket," our ProEdit generates much clearer and more noticeable plaid patterns than all baselines.

The experimental results on the ScanNet++ dataset are shown in Fig. 4. With subtask decomposition and progressive editing, our ProEdit achieves high-quality results that are comparable to and even outperform the baseline ConsistDreamer [9], which incorporates three complicated add-ons for ensuring 3D consistency. This shows that our simple progression is more effective in reducing inconsistency – through reducing the size of FOS – than explicit 3D consistency-enforcing components.

| Method | USO↑ | US3D↑ | GPT↑ | CTIDS↑ | CDC↑ | Running Time↓ |
|---|---|---|---|---|---|---|
| IN2N [5] | 51.35 | 65.45 | 45.32 | 0.0773 | 0.3260 | **0.5-1h** |
| ConsistDreamer [9] | 68.65 | 75.23 | 74.40 | **0.0912** | **0.3912** | 12-24h |
| ProEdit (Ours) | **87.96** | **80.23** | **81.00** | 0.0844 | 0.3833 | 1-4h |

Table 1: Our ProEdit significantly outperforms baselines in USO, US3D, and GPT metrics, and achieves comparable CLIP metrics to sophisticated ConsistDreamer with only 1/3 of its running time.

| Method | USO↑ | US3D↑ | USP↑ | GPT↑ | CTIDS↑ | CDC↑ |
|---|---|---|---|---|---|---|
| ProEdit (ND) | 68.46 | 61.72 | 60.73 | 72.87 | 0.0671 | 0.2902 |
| ProEdit (Full) | **92.70** | **90.48** | **88.72** | **82.80** | **0.0844** | **0.3833** |

Table 2: **Ablation study of our "no subtask decomposition (ND)" variant** shows that our full ProEdit significantly outperforms the "ND" variant across all metrics, validating that progression is crucial to achieving high-quality editing results.

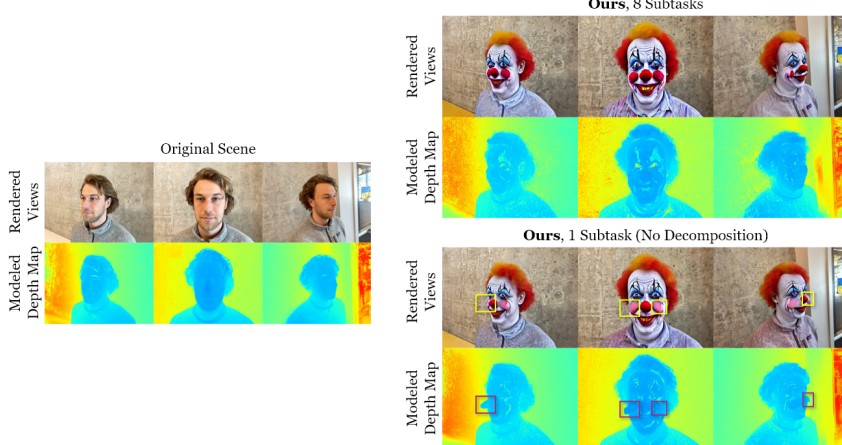

Figure 6: **Ablation study of our "no subtask decomposition" variant** shows that removing subtask decomposition results in unreasonable geometry, particularly near the cheek area (indicated by the bounding boxes). This validates the importance of subtask decomposition in achieving high-quality editing in our framework. "Modeled depth map" is the depths modeled by the scene representation.

We also conduct comparison experiments on two outdoor scenes: "Bear" from IN2N [5] and "Floating Tree" from NeRFStudio [44], as shown in Fig. 5. In the "grizzly bear" task, our ProEdit generates similar fur textures as ConsistDreamer, both of which are much clearer than IN2N, with the added advantage of aggressivity control in our model. Notably, our ProEdit achieves comparable editing quality at only 1/4 to 1/6 of ConsistDreamer's running time and with fewer GPUs. In the "snow" task, our ProEdit also delivers high-quality editing results, generating snow on the ground and making the sky whiter, while the baseline IN2N creates a blurred ground and leaves. In the "autumn" task, our ProEdit demonstrates its aggressivity control by adjusting the color intensity of the leaves. These results highlight the effectiveness of our approach for outdoor scenes as well.

In addition, our ProEdit shows the capability to control and categorize the aggressivity level of editing tasks. By selecting the edited scene from an intermediate subtask, we can obtain scenes with varying levels of aggressivity – namely, medium and high aggressivity, as shown in Figs. 3, 4, and 5 – with noticeable discrepancies. For example, in the medium-aggressivity version of the "Tolkien Elf" editing in Fig. 3, only the eye color and ear shape are modified, while in the high-aggressivity version, not only are the ears lengthened, but the hair is also colored red, and the face is thinned. These results underscore the unique strength of our ProEdit in controlling editing aggressivity.

Additional qualitative results are shown on our project page.

**Quantitative Results.** Table 1 presents quantitative comparisons. ProEdit consistently outperforms IN2N by a large margin. It also significantly surpasses the strong baseline ConsistDreamer in two overall quality metrics and the user study-based 3D consistency metric, while achieving comparable performance on CLIP-based metrics – all with only 1/3 of ConsistDreamer's running time.

**Ablation Study.** To validate the necessity of our subtask decomposition, we conduct experiments on a variant of ProEdit using only one subtask ($n = 1$, $r_0 = 0$, $r_1 = 1$), effectively disabling decomposition (referred to as "ND"). Qualitative results are shown in Fig. 6. Without subtask decomposition, the variant generates unrealistically long cheeks to accommodate inconsistencies in cheek decorations across views, resulting in blurred cheek textures in the rendered output due to the large FOS of the editing task. In contrast, our full ProEdit achieves bright, clear results with precise and realistic geometry. Quantitative results are shown in Table 2. For this comparison, we conducted a new user study involving 41 participants, including an additional User Study of Shape Plausibility ("USP") metric: we provide participants with the modeled depth maps, similar to those in Fig. 6, along with the rendered RGB images. We then ask them to evaluate whether the shapes are realistic and match the rendered images. The "ND" variant performs significantly worse than our full method on all user study metrics, further underscoring the effectiveness of our subtask decomposition. These results collectively demonstrate that reducing FOS through subtask decomposition is crucial to our high-quality results.

## 5  Discussion

**3D Consistency Add-Ons.** Different from our subtask decomposition strategy, 3D consistency add-ons, such as distillation losses, consistency-inducing components, and specific training procedures, offer an alternative way to control and reduce FOS. Although our framework achieves high-quality editing without them, combining it with these 3D consistency add-ons can leverage the strengths of both approaches, potentially reducing the number of required subtasks and enhancing editing quality.

**Limitations.** Our ProEdit is a distillation-guided framework from 2D diffusion, similar to all baselines. Therefore, its editing capability is constrained by the underlying diffusion model. If the diffusion model does not support applying a specific editing instruction on most views of a scene, our ProEdit will also be unable to do so. Additionally, ProEdit relies on 3DGS for efficient training, which NeRF-based representations do not support; consequently, it inherits certain limitations of 3DGS, including limited suitability for unbounded outdoor scenes. Finally, ProEdit may still encounter the multi-face or Janus problems, as the 2D diffusion model lacks 3D awareness.

**Future Directions.** There are many promising directions to explore in subtask decomposition beyond the interpolation-based strategy introduced in this paper. One potential way is to explicitly construct intermediate subtasks using semantic guidance. For example, applying "Turn him into a bald person" before "Make him wear a hat" could lead to a more free-form hat independent of the hair, with such intermediate instructions generated by large language models. Another alternative avenue involves leveraging video generation models to "animate" the transition from the original scene to the edited scene, treating this animation process as a series of subtasks. Doing so will enable ProEdit to function as a 3D scene animator, generating high-quality 4D (dynamic 3D) scenes. Additionally, the progressive framework of ProEdit can be potentially applied to scene generation.

**Potential Societal Impacts.** The positive societal impacts of our ProEdit include (1) the development of consumer-grade 3D scene editing products and applications, facilitated by advancements in 3D structured-light scanners for mobile phones and virtual reality (VR) and augmented reality (AR); and (2) the transformation of high-quality 3D and 4D (dynamic 3D) scene creation through the editing of existing high-resolution scenes. On the other hand, as our framework is based on generative models, it is crucial to address potential ethical and safety concerns, including risks of producing biased results and the possibility of misuse for illegal activities.

## 6  Conclusion

This paper proposes ProEdit, a novel 3D scene editing framework that decomposes the editing task into subtasks and performs them progressively. Our method targets the fundamental cause of inconsistency – the large feasible output space of the diffusion model with respect to an editing task. Extensive experiments show that our ProEdit produces high-quality editing results characterized by bright colors, sharp and detailed textures, and precise geometric structures across various scenes and editing tasks. Our method further enables a novel controllability over the aggressivity of the editing task, by allowing users to select which subtasks to execute. We hope that our ProEdit will inspire exciting applications and new research directions in 3D scene editing and generation.

# Acknowledgments

This work was supported in part by NSF Grant 2106825, NIFA Award 2020-67021-32799, the IBM-Illinois Discovery Accelerator Institute, the Toyota Research Institute, and the Jump ARCHES endowment through the Health Care Engineering Systems Center at Illinois and the OSF Foundation. This work used computational resources on NCSA Delta through allocations CIS220014 and CIS230012 from the Advanced Cyberinfrastructure Coordination Ecosystem: Services & Support (ACCESS) program, and on TACC Frontera through the National Artificial Intelligence Research Resource (NAIRR) Pilot.

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

# Appendix

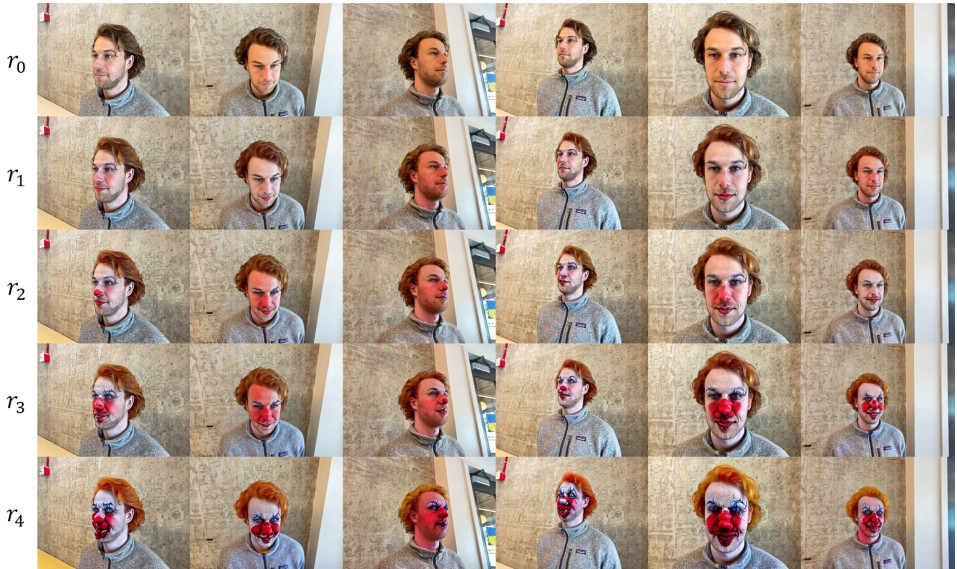

Figure A: The visualization of per-view edited results (where each view is edited separately with IP2P [4]) shows that with the increment of the subtask ratio $r$, the multi-view inconsistency also increases, leading to greater editing difficulty.

## A   Editing Difficulty w.r.t. Subtask Ratio $r$

We provide a visualization of per-view edited results (*i.e.*, each image is edited *individually* with IP2P [4]) with respect to different $r$'s, as shown in Fig. A. The multi-view inconsistency situations are as follows:

- $r_0$: All views remain identical to the original view, resulting in perfect consistency.
- $r_1$: The face begins to turn white, with the only inconsistency being the different degrees of color change.
- $r_2$: Some areas of the face become red, introducing a new inconsistency in different locations of the red parts.
- $r_3$: Additional areas of the face change color, and the nose alters shape, leading to increased inconsistencies in color distribution and nose shape.
- $r_4$: The final edited results exhibit various inconsistencies across all parts, even including changes to hair color.

This visualization shows that the editing inconsistency and difficulty increase as $r$ rises.

## B   Additional Subtasks $r_0$ and $r_n$

Our ProEdit framework is designed to accept *any* scene representation for input and output, including NeRFs [1] and conventional 3DGS [2]. However, our editing process requires the scene representation to be our adaptive 3DGS (Sec. 3.3), which is optimized for progressive editing.

Therefore, the additional subtasks $r_0$ and $r_n$ represent the input and output states where the scene is in other representations. The corresponding subtasks $s_0 = S(s_{\text{input}}, r_0)$ and $s_{\text{output}} = S(s_n, r_n)$ are for the conversion between these other scene representations and our adaptive 3DGS (*i.e.*, re-reconstructions). Within such re-reconstructions, the diffusion model acts as a simple refiner for the per-view images, which preserves most appearances while refining defects or abnormalities and compensating for minor inadequacies in the edited areas.

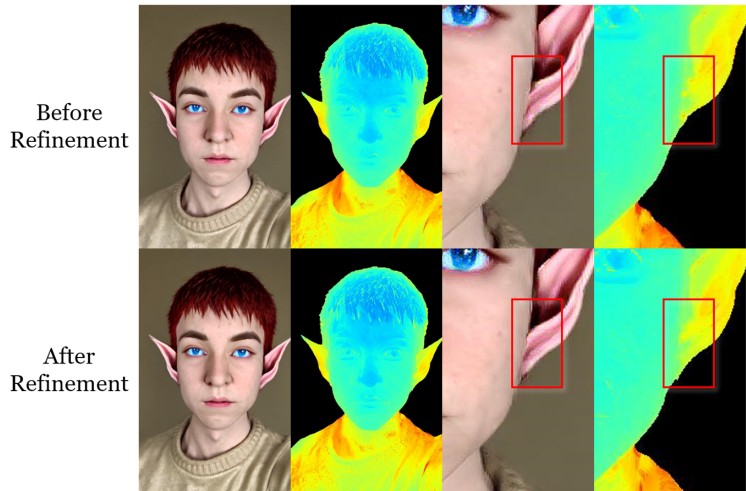

Before Refinement

After Refinement

Figure B: The additional subtask $r_n$ does not significantly change the overall appearance, but it brings slight improvements on geometric structure.

We present a visualization of the results before and after the refinement from the additional subtask $r_n$ in Fig. B. The depth maps, modeled by 3DGS, are segmented to emphasize the foreground. We can observe that the two images exhibit similar appearances, but the refined version demonstrates more precise geometry and detail near the ear. This shows that while the additional subtask $r_n$ does not lead to significant changes or improvements in overall appearance, it provides minor refinements to the edited results.

## C  Consistency and Non-Linearity of Subtasks

In our method, we employ adaptive task decomposition (Sec. 3.2) to reduce the difficulty and inconsistency of each subtask. We approximate the difficulty by measuring the difference between the original and edited images, and design an adaptive subtask decomposition upon this difference. Even if the instruction encoder $E(\cdot)$ is non-linear, this method enables us to achieve subtask decomposition with reduced difficulty, ensuring that the difficulty of each subtask does not exceed $d_{\text{threshold}}$, a preset threshold for subtask difficulty.

As our method decomposes one editing task into multiple subtasks, we need to solve more editing (sub-)tasks in total. While completing all these subtasks may require a longer total running time, each subtask is simpler and therefore faster to achieve than performing the entire editing task in one step. This trade-off allows us to significantly improve performance while gaining control over editing aggressivity. Notably, our ProEdit is considerably more efficient than the current state-of-the-art, ConsistDreamer, as detailed in Table 1.

## D  Gaussian Creation Strategy

Our Gaussian creation strategy regulates the growth speed of the Gaussians. Specifically, if we culled $n$ Gaussians in the previous step, we will only allow $t(n)$ Gaussians to be created at the current step, where $t(n)$ represents a threshold function based on $n$ and the total number of Gaussians.

This control strategy for Gaussian creation (1) prevents the generation of excessive Gaussians for minimally inconsistent multi-view images, and (2) concentrates Gaussian generation in high-frequency regions of the scene, improving the overall results.

## E  GPT Evaluation Score

For the "GPT evaluation score" metric, we provide GPT-4o [49] with the original video, the editing prompt, and the videos generated by three methods all together with random names and in random

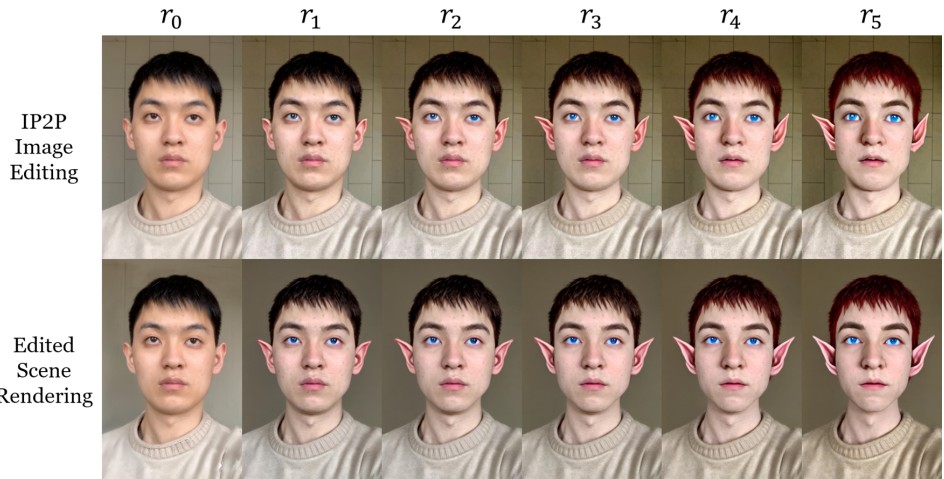

Figure C: The edited 3D scene at each decomposed subtask aligns well with the corresponding interpolated tasks edited by IP2P [4] for the image.

order to ensure unbiased scoring. GPT-4o then evaluates each video on a scale of 1 to 100, considering (1) editing completeness and accuracy, (2) preservation of the original content, (3) 3D consistency, and (4) visual appearance. The scores for all baselines are returned as a JSON array. We repeat this process multiple times and report the average score.

Our GPT score functions as a Monte-Carlo implementation of the recently proposed VQAScore [50], a metric that leverages vision-language model evaluation of generated images and has been shown to outperform CLIP-based [48] scores. Given the advanced capabilities of vision-language models to evaluate various relevant aspects, this VQA-based metric, along with our GPT score, offers a more comprehensive automated quantitative measurement than CLIP-based scores.

## F   Semantic Meaning of Subtasks and Alignment with IP2P Editing

In our method, the semantic meaning can be interpreted as "how IP2P [4] behaves with the interpolated embedding." Although it is difficult to explicitly define text instructions for the interpolated embedding, we can still visualize them with IP2P's editing results corresponding to the interpolated embeddings.

We provide a visualization illustrating the alignment of the edited scene in Fig. C. From the first row, we can roughly interpret each subtask's goal. For example, $r_2$ suggests "give him blue eyes and pointy ears; make the background slightly green," while $r_4$ indicates "make his eyes completely blue, his hair red, and his face slightly thinner; make the background dark green." Comparing the two rows, we can observe that the edited scene roughly matches the appearance and effect of IP2P image editing, particularly in hair color. This reveals the semantic alignment for each subtask.

Although our task decomposition uses a linear interpolation of embeddings, our method is agnostic to their exact semantic meaning. Our key insight is to decompose a difficult task into several easier tasks to reduce the inconsistency during distillation (Sec. 3.2). Instead of focusing on the semantic meaning of the interpolated embeddings, we emphasize whether a selected interpolation point effectively decreases difficulty and inconsistency. Therefore, our difficulty metric for adaptive subtask decomposition is designed based on the approximated task difficulty, rather than semantic differences.

