# OpenReview forum: "ProEdit: Simple Progression is All You Need for High-Quality 3D Scene Editing"
_NeurIPS.cc/2024/Conference — NeurIPS 2024 poster_

### Official Review · Reviewer_PcMD · 2024-06-14

**Soundness:** 3
**Presentation:** 3
**Contribution:** 3
**Rating:** 6
**Confidence:** 4

**Summary:**

This paper presents a task decomposition method that achieves more robust 3D scene editings. The main contribution is the concept of decomposition of the desired task (represented by a prompt) and the adaptive 3D Gaussian Splatting training process. The edited appearance and geometry using decomposed tasks are better than the results generated by previous methods.

**Strengths:**

The strengths of this paper are:
- The idea of decomposing the task is interesting and useful.
- The results demonstrated in the paper is convincing to support the claim.

**Weaknesses:**

The weaknesses of this paper are:
- The simple linear decomposition scheme do not provide any semantic meaning, which is a bit hard to evaluate the result of each subtasks.
- Moreover, there is no results (except fig. 1) of each subtask.

**Questions:**

- It is unclear can the method do more object manipulation instead of just appearance editings. For example, it would be nice to show more examples for ScanNet++ scenes that involves moving objects or adding objects. Current results are mostly just change the appearance.
- I think it is important to show and evaluate the results of each subtask, instead of just showing the final results in the main paper. (e.g., use a more complex task and test the alignment between the results of each subtask and the human task decomposition results). I understand this is still a limitation of simple linear interpolation scheme, but the results will be very meaningful and helpful imho for future research.

**Limitations:**

I already discussed the main limitation, i.e., the task decomposition in the previous section.

---

> ### Author Rebuttal · Authors · 2024-08-07
>
> ### W1. The semantic meaning of decomposition and evaluation of subtasks
> - In our method, the semantic meaning can be interpreted as "how IP2P acts with such interpolated embedding." Though we cannot write down the text instructions corresponding to the interpolated embedding, we can still visualize it with the IP2P's editing results w.r.t. the interpolated embeddings. We provide a visualization of the alignment between the edited scene in **FigPDF.E**.
>   - From the first row, we can roughly interpret each subtask's goal. For example, $r_2$ roughly indicates, "give him blue eyes and pointy ears; make the background slightly green," while $r_4$ indicates, "make his eyes completely blue, his hair red, and his face slightly thinner; make the background dark green."
>   - Comparing the two rows, we can observe that the edited scene roughly matches the appearance/effect of the IP2P image editing, as especially shown in hair color. This shows a subtask semantic alignment for each subtask.
> - Though our task decomposition is based on a linear interpolation of embeddings, our method does not depend on, and is actually agnostic to, its exact semantic meaning.
>   - Our key insight is to decompose a difficult task into several easier tasks to reduce the inconsistency in the distillation (Sec.3.2). Instead of focusing on the semantic meaning of the interpolated embeddings, we focus more on whether and how selecting an interpolation point can sufficiently decrease the difficulty and inconsistency.
>   - Therefore, our difficulty metric for adaptive subtask decomposition (L159) is designed based on the approximated task difficulty, instead of the difference of semantic meanings.
>
> ### W2/Q2. Results of each subtask
> - We have provided the results of each subtask in our supplementary video at 0:03. Please refer to it.
>   - As shown at 0:03 of the supplementary video, a latter subtask results in more aggressive editing (e.g., redder hair, thinner face, and larger ears in the "Elf" editing task).
> - We also provide a visualization of the alignment between the edited scene and IP2P's behavior on each subtask. Please refer to the response to W1 above and **FigPDF.E**.
> - We will provide more results in the main paper in the revision.
>
> ### Q1. Tasks about moving or adding objects in the scenes
> - We would like to clarify that, following previous works IN2N, ViCA-NeRF, ConsistDreamer, etc., our paper focuses on a framework to perform instruction-guided editing, that distills the editing signals from an existing pre-trained 2D diffusion model. We would like to humbly point out that the operation of moving or adding objects is a challenging task that is not yet solved by any of the baselines, and is out of the scope of this paper.
>   - The method we propose is a distillation-guided instruction-guided pipeline. Therefore, the editing capability of our framework, as well as the existing baselines, is all distilled from the 2D diffusion model, which needs to support object movement/creation guided by instruction. However, currently, most 2D diffusion models do not well-support an instruction-guided object moving, and multiple views may even require different instructions to perform the editing (e.g., "left" should be changed to "right" in an opposite view, different visible reference objects, etc.).
>   - Designing such a 2D diffusion model and/or a format of 3D-consistent editing instruction is out of the scope of this paper. We leave this task as an interesting future work. However, the idea of progressive editing is general, and can be potentially applied to such a task once we have an applicable 2D diffusion model for this task with appropriate instruction conditions.

---

### Official Review · Reviewer_49ym · 2024-06-30

**Soundness:** 2
**Presentation:** 3
**Contribution:** 3
**Rating:** 6
**Confidence:** 4

**Summary:**

This paper presents ProgressEditor, which decomposes the 3D scene editing task into multiple subtasks and progressively modifies the scene which is represented by 3D Gaussians. The subtask decomposition is defined as the linear interpolation of the encoding of the editing prompt. Given the editing instruction, the proposed approach recursively searches for the proper subtask decomposition, so that the difficulty of each subtask is uniformly distributed. Then it progressively completes the subtasks with the proposed adaptive Gaussian creation strategy and finally obtains the edited scene.

**Strengths:**

The strengths of this paper include:
(1) It proposes a novel idea that decomposes the 3D editing task into multiple subtasks and progressively completes the editing.
(2) It provides an adaptive 3DGS tailored to the progressive 3D editing framework, which is able to refine the 3DGS more efficiently.
(3) The proposed approach generates high-quality editing results with clear texture and precise geometry.

**Weaknesses:**

The weaknesses include:
(1) The experimental evaluation is insufficient to validate the effectiveness of the proposed approach with the lack of quantitative assessment.
(2) It lacks the ablation study on some technical designs described in the method section. Please see my questions below.

**Questions:**

The key of the proposed approach is to decompose the 3D scene editing tasks to alleviate the multi-view inconsistency problem during the distillation process. However, what about the consistency of each subtask? The instruction encoding is not necessarily linear. So the inconsistency still exists in each subtask. Although each subtask should have less inconsistency compared to the original task, would the progressive editing (multi-stage editing) introduce additional burden to the process?

Line 194-196 describe the subtask scheduling, where it adds additional subtasks r_0 and r_n. There should be an ablation study to validate the necessity of this setting. There should be more technical details about the Gaussian creation strategy described in lines 229-231.

It is difficult to measure the advantage of the proposed method compared to the other alternatives. It seems the generated results of the proposed method preserve the feature of the person better in Fig.3. But it’s hard to say that it generates the results with more geometry editing (line 283). A quantitative evaluation should be reported to validate the effectiveness of the proposed method. In addition, a comparison of the time cost of each method should also be presented.

**Limitations:**

This paper includes a discussion of the limitations of the proposed approach. It's better to present some failure cases for a better understanding.

---

> ### Author Rebuttal · Authors · 2024-08-07
>
> ### W1. Quantitative assessment
> - Please refer to the "Quantitative Evaluation" in the **global author rebuttal**. Thank you.
>
> ### W2/Q2.2. Additional subtasks $r_0$ and $r_n$
> - Our framework is designed in a setting, where the input and output format of the scene can be in *any* scene representation (e.g., NeRFs, conventional 3DGS, etc.). However, our editing procedure requires the scene representation to be our *Adaptive* 3DGS, which is tailored to progressive editing.
>   - Therefore, the additional subtasks $r_0$ and $r_n$ represent the input and output states where the scene is in other representations. The corresponding subtasks $s_0 = S(s_\mathrm{input}, r_0)$ and $s_\mathrm{output} = S(s_n, r_n)$ are for the conversion between other scene representations and our adaptive 3DGS (i.e., re-reconstructions).
>   - Within these re-reconstructions, the diffusion model works as a simple refiner of the per-view images, which preserves most of the appearances and only refines some defects or abnormalities, and may also compensates for some minor non-sufficient edited parts.
> - We provide a visualization of results before and after the refinement of addition subtask $r_n$ in **FigPDF.D**, where the depth maps are the 3DGS-modeled depth, segmented to emphasize the foreground. We can observe that the two images have very similar appearances, while the refined version has a more precise geometry and appearance near the ear part. This shows that the additional $r_n$ can make minor refinements to the edited results but will not significantly change or improve the appearance.
>
> ### Q1. The consistency and non-linearity of subtasks
>
> - In our method, we use *adaptive* task decomposition to reduce the difficulty or inconsistency of each subtask. As mentioned in L150-L180, we approximate the difficulty with the difference between original and edited images, and design an adaptive subtask decomposition upon this.
>   - With this method, even if the instruction encoder is not linear, we can still obtain a subtask decomposition with reduced difficulty in between, which is no larger than $\mathrm{d}_\mathrm{threshold}$ (L173), a preset threshold for subtask difficulty.
>   -  As our method decomposes one editing task into multiple subtasks, we have to solve more editing (sub-)tasks in total. Though we may still need a longer running time to complete all these subtasks, each decomposed subtask is simpler to achieve and, therefore, takes a shorter time than completing a full editing task, and we can significantly improve the performance and gain aggressivity controllability with this trade-off. Notably, our ProgressEditor is significantly more efficient than current state-of-the-art ConsistDreamer, as detailed in the reply to Q3.2 below.
>
> ### Q2.1 Gaussian creation strategy in L229-L231
> - This strategy is about controlling the growth speed of the Gaussians. More specifically, if we culled $n$ Gaussians in the previous step, we will only allow $t(n)$ Gaussians to be created at this step, where $t(n)$ is the threshold scheduling w.r.t. $n$ and the total number of Gaussians.
> - With this controlling strategy, the Gaussian creation (1) will not generate too many Gaussians for slightly inconsistent multi-view images, and (2) will create more Gaussians at the high-frequency parts of the scene, which improves the results.
> - We will add these details in the revision.
>
> ### Q3.1. Where our results have more geometry editing
>
> - Our ProgressEditor performs the editing with more geometry editing, as shown in the following editing cases:
>   - In the "Tolkien Elf" task of Fangzhou scene in Fig.3, ours w/ high aggressivity makes the shape of the face thinner, while most baselines tend to preserve the original shape.
>   - In the "Lord Voldemort" task of Fangzhou scene in Fig.3, ours generates a face with more wrinkles and also edits the neck part.
>   - In the "Clown" task of the Face scene in Fig.3, the clown edited by ours is smiling more aggressively.
>
> ### Q3.2.  Comparison of time cost
>
> - As we are using a dual-GPU training strategy (L233), each subtask takes only slightly longer than training a 3DGS representation from scratch (10-15 minutes). Therefore,  the whole editing process with 4 subtasks takes 1-2 hours, and the ones with 8 subtasks take 3-4 hours.
> - Compared with baselines, ConsistDreamer takes 12-24 hours according to their paper; other baselines like IN2N may take comparable or shorter time than ours, but can only achieve lower-quality editing results with significantly worse 3D consistency.

---

> > ### Comment · Reviewer_49ym · 2024-08-09
> >
> > I thank the authors for their response. The response clarifies the details about "additional subtasks" and "consistency and non-linearity of subtasks", and "Gaussian creation strategy". My only remaining concern is the "ablation study on some technical designs". Specifically, the authors have already shown the "no subtask decomposition" variant in Fig.5 of the paper. Is there any quantitative evaluation (like the newly added comparison evaluation) to validate the effectiveness of the subtask decomposition, which is the main novelty of the proposed approach? I'm happy to increase my rating given any quantitative evidence about this ablation study.

---

> ### Author Response · Authors · 2024-08-10
>
> We sincerely thank the reviewer for acknowledging our clarifications, and we are glad that our response has addressed most of the reviewer’s concerns. We thank the reviewer for the follow-up question and address the remaining concern here.
>
> ### D1. Quantitative ablation study
>
> - We provide the quantitative comparison between our full method and the “No Decomposition” (“ND”) variant, as shown in the table below.
>
>     | Variant | GPT↑ | CTIDS↑ | CDC↑ |
>     | :---- | :---- | :---- | :---- |
>     | Ours ND | 72.87 | 0.0671 | 0.2902 |
>     | Ours Full | **82.80** | **0.0844** | **0.3833** |
>
>   - Note: To promptly address the reviewer's question, here we primarily focus on GPT and CLIP-based metrics, which we think are sufficient to validate the effectiveness of our method. We leave the user study later, as it requires additional time to gather user responses.
>   - In addition, the GPT score of our full method presented here is not directly comparable to that in the global author rebuttal. In this case, we compare our full method with the “ND” variant, whereas in the global author rebuttal, we compared our full method with the IN2N and ConsistDreamer baselines.
>
> - Without subtask decomposition, the “ND” variant directly exposes the 3DGS to highly inconsistent edited multi-view images. This makes the 3DGS overfit to such inconsistent images with view-dependent effects and finally leads to consistently lower metrics. Together with the visualization (e.g., Fig.5), this quantitative evaluation validates the effectiveness of our proposed subtask decomposition.
>
> If the reviewer has any follow-up questions, we are happy to discuss them.

---

> ### Author Response · Authors · 2024-08-11
>
> ### D1 (Continued). Quantitative ablation study \- User study
>
> - Following our earlier response, we have now obtained the results from our user study involving 41 participants. The results are shown in the table below, with the following metrics: user study of overall quality ("USO"), user study of 3D consistency ("US3D"), and user study of shape plausibility ("USP", detailed as below).
>
>     | Variant | USO↑ | US3D↑ | USP↑ |
>     | :---- | :---- | :---- | :---- |
>     | Ours ND | 68.46 | 61.72 | 60.73 |
>     | Ours Full | **92.70** | **90.48** | **88.72** |
>
>   - Please note that similar to the GPT scores, the USO and US3D metrics here are not directly comparable with those in our global rebuttal.
>   - For this user study, we further evaluate shape plausibility (USP) as an additional metric. We provide participants with the modeled depth maps, similar to those in Fig.5, along with the rendered RGB images. We then ask them to evaluate whether the shapes are reasonable and match the rendered images.
>
> - Consistent with the conclusion in our earlier response, the "ND" variant performs significantly worse than our full method under the user study in all metrics. This further validates the effectiveness of our proposed subtask decomposition.

---

> > ### Comment · Reviewer_49ym · 2024-08-12
> >
> > Thank you for making the quantitative evaluation! The numbers validate the effectiveness of the subtask decomposition idea.
> >
> > So I have improved my score to weak accept.

---

> > > ### Author Response · Authors · 2024-08-12
> > >
> > > We sincerely thank the reviewer for the positive feedback and for raising the score. Your constructive comments and suggestions have been invaluable in improving the paper.

---

### Official Review · Reviewer_Mz9K · 2024-07-02

**Soundness:** 3
**Presentation:** 3
**Contribution:** 3
**Rating:** 6
**Confidence:** 5

**Summary:**

This work focuses on instruction-based 3D scene editing. It proposes a progressive editing framework by decomposing the complex editing task into different subtasks based on the difficulties. In this way, it could ensure multi-view consistency in each easy subtask and finally obtain consistent editing for the whole task.

**Strengths:**

1.  The idea of decomposing a difficult task into several easy subtasks is interesting and makes sense. This can avoid inconsistent multiview edits based on a complex instruction.

2.  The figure of pipeline clearly illustrates the methodology and motivation. Based on the visualization results, the proposed method demonstrates superior editing effects compared to the baseline methods.

**Weaknesses:**

1. I have some doubts about the main technique of the article. This work defines sub-tasks of different difficulties by weighting instruction prompts and empty prompts with r. I am uncertain whether the editing difficulty is sufficiently sensitive to the weight r. The authors should provide 2D multiview editing results for different r to illustrate that as r increases, the inconsistency across multiple views also increases, indicating a rise in editing difficulty.

2. The illustrations in Fig.3 are not clear. I don't know which row/column corresponds to which method. Please clarify this in the response.

3. This paper does not provide a quantitative comparison with baselines.

4. Most of the results are based on human faces. The edits conducted on Scannet scenes are style transfer, which does not require high levels of consistency. It would be better to give more visualizations of outdoor scenes such as the 'bear' and 'garden', etc.

**Questions:**

Please refer to the weakness. My major concern is the reasonability of defining the difficulties based on the weight r.

**Limitations:**

The authors have discussed the possible limitations and social impacts.

---

> ### Author Rebuttal · Authors · 2024-08-07
>
> ### W1/Q1. About the editing difficulty w.r.t. the weight $r$.
> - We provide a visualization of per-view edited results (i.e., each image is edited *individually* with IP2P) w.r.t. different $r$'s as **FigPDF.A**. The multi-view inconsistency situations are as follows:
>   - $r_0$: All the views are the same as the original view, so it is perfectly consistent.
>   - $r_1$: The face begins to turn white. The only inconsistency is the different degrees of changing color.
>   - $r_2$: Some parts of the face become red, with a new inconsistency in different locations of red parts.
>   - $r_3$: More parts of the face change the color, and the nose changes the shape. More inconsistency emerges, including color distribution and nose shape.
>   - $r_4$: The final edited results with various inconsistencies in all parts, even including the hair color.
>   - This visualization shows that the editing inconsistency and difficulty increase when $r$ increases.
> - In Sec. 3.2, we also show some analysis about the increment of difficulty when $r$ increases.
>   - As mentioned in L138, the task difficulty is proportional to the size of the feasible output space (FOS, the set of possible scenes that can be regarded as the edited result of the editing task).
>   - When $r=0$, the task is just "keeping original," and the FOS only contains the original scene. With the increment of $r$, the task becomes more aggressive, i.e., far from "keeping original," and brings a more significant change to the scene.
>   - As mentioned in L153, "intuitively, an editing task that brings a significant change typically has more degrees of freedom to apply such a change, leading to a larger FOS," and therefore, higher difficulty and more inconsistency.
>   - With subtask decomposition, we only need to solve subtasks from $r_{i-1}$ to $r_{i}$, which has a much smaller FOS compared with directly solving the subtask $r_{i}$ (from $r_0=0$), and, therefore, is much easier.
>
> ### W2. The illustrations in Fig.3
> - Please refer to the maps in **FigPDF.B** to help understand the organization of Fig.3.
> - The upper part of Fig.3 contains the results of the Fangzhou scene. There are two sub-tables on the left and right. In each of the sub-tables, each row corresponds to a method (baseline or ours), and each two-column group represents an editing task (e.g., "Turn him into the Tolkien Elf").
> - The lower part of Fig.3 contains two parts.
>   - The first row represents the original scene, and the editing results of the task "Turn him into a clown."
>   - In the first 6 columns of the table below, each row corresponds to a method, and each two-column group represents an editing task.
>   - The last 4 columns show the crucial editing task "Give him a plaid jacket." The two rows on the top-left continue the same rows of the first 6 columns, which correspond to baselines "IN2N" and "ConsistDreamer," and the two rows on the top-right are two other baselines "PDS" and "EN2N." The two rows on the bottom also continue the same rows of the first 6 columns, which are "ours" under two settings.
> - We apologize for the confusion caused by the layout of Fig.3. As different baselines have different publicly available results (e.g., some methods provide code for re-production, while the others can only be compared by referring to the images provided in their papers), we have to organize Fig.3 in this way to show all of them concisely. We will revise the caption to include a detailed explanation of its organization in the revision.
>
> ### W3. Quantitative comparison
> - Please refer to the "Quantitative Evaluation" in the **global author rebuttal**. Thank you.
>
> ### W4. Results of outdoor scenes
> - In our original submission, we emphasized the results of the scenes which were mostly covered by the baselines, for a thorough comparison.
> - We thank the reviewer for the suggestion and here we provide the results of two outdoor scenes: Bear from IN2N and Floating Tree from NeRFStudio, in **FigPDF.C**. As ConsistDreamer was not evaluated on the Floating Tree scene, we only compare with IN2N in the editing tasks of this scene.
>   - In the "grizzly bear" task, our ProgressEditor generates similar fur textures as ConsistDreamer, both of which are much clearer than IN2N, and ours also supports aggressivity control. Notably, our ProgressEditor achieves comparable editing results with only 1/4 to 1/6 running time of ConsistDreamer with fewer GPUs.
>   - In the "snow" task, our ProgressEditor can also provide high-quality editing results by generating snow on the floor and making the sky whiter, while the baseline IN2N generates blurred floor and leaves. In the "autumn" task, our ProgressEditor also shows the aggressivity control ability by controlling the color of the leaves.
>   - These results demonstrate that our approach is effective for outdoor scenes as well. We will add these and more results of outdoor scenes in the revision.
> - We would also like to clarify that high levels of consistency are also crucial in style transfer tasks, especially for large-scale scenes in ScanNet++. For such scenes, each object may occur in many different views from various viewing directions, and inconsistent multi-view editing results in more blurry and gloomy colors after averaging. As shown in the visualization figures in ConsistDreamer's paper, namely Fig.1 (Van Gogh painting) and Fig.B.4 (Ablation Task (B)/(D)), inconsistency results in gloomy colors and blurred textures in style transfer tasks, especially in ScanNet++ scenes.

---

> > ### Comment · Reviewer_Mz9K · 2024-08-09
> >
> > Thanks for your responses and clarifications. These responses have solved most of my concerns and questions. I finally decided to improve my score to weak accept.

---

> > > ### Author Response · Authors · 2024-08-10
> > >
> > > We sincerely thank the reviewer for the positive feedback and for increasing the score. If the reviewer has any follow-up questions, we are happy to discuss them.

---

### Author Rebuttal · Authors · 2024-08-07

We thank the reviewers for their constructive and insightful comments:

- We propose an "interesting, novel, and reasonable" idea (Mz9K, 49ym, PcMD) to solve the instruction-guided 3D editing task, in a well-presented and illustrated way with clearly stated motivations (Mz9K).
- The proposed method generates "high-quality editing results with clear texture and precise geometry" (49ym), which are  "superior" compared to the baselines (Mz9K) and "convincing to support" the effectiveness of our method (PcMD).

We address all the reviewers' concerns in each reply. We also provide the following visualizations in our PDF content:
- **FigPDF.A**: Per-view edited results of different subtasks, as a visualization of the difficulty at different $r$'s. (Mz9K)
- **FigPDF.B**: Map of organization of Fig.3. (Mz9K)
- **FigPDF.C**: Results of outdoor scenes. (Mz9K)
- **FigPDF.D**: Visualization of results before and after additional subtask $r_n$. (49ym)
- **FigPDF.E**: Visualization of the alignment between subtasks and edited scene. (PcMD)

We also reply to some commonly asked questions here.

### Mz9K-W3 / 49ym-W1. Quantitative Evaluation

- In our original submission, we primarily focused on the extensive qualitative comparisons to show the advantages of our high-quality editing results, as it still remains an open question to design a metric to evaluate the 3D editing results in a fair and complete manner.
- We provide the quantitative assessment with the following metrics: user study of overall quality ("USO"), user study of 3D consistency ("US3D"), GPT evaluation score ("GPT"), CLIP Text-Image Direction Similarity ("CTIDS"), and CLIP Direction Consistency ("CDC"). The user study was conducted with 26 participants. The GPT score is detailed below, and the CLIP-based scores are from IN2N's paper. The results are shown in the table below.

  |Method|USO↑|US3D↑|GPT↑|CTIDS↑|CDC↑|Running Time↓|
  |:-|:-:|:-:|:-:|:-:|:-:|:-:|
  |IN2N           |  51.35  |  65.45  |  45.32  |  0.0773  |  0.3260  |**0.5-1h**|
  |ConsistDreamer | _68.65_ | _75.23_ | _74.40_ |**0.0912**|**0.3912**|  12-24h  |
  |ProgressEditor (Ours) |**87.96**|**80.23**|**81.00**| _0.0844_ | _0.3833_ |  _1-4h_  |

  - From the table above, we observe that our ProgressEditor consistently outperforms IN2N with large margins. Our ProgressEditor also significantly outperforms the strong baseline ConsistDreamer in two overall-quality metrics and the user study-based 3D consistency metric, while achieving comparable CLIP-based metrics, with a running time of only 1/3 of that of ConsistDreamer.
  - "GPT score": We provide GPT-4o with the original video, the editing prompt, and the video generated by three methods all together with random names and in random order (to enforce a consistent scoring mechanism across methods). Then, we ask it to provide a score between 1 and 100 for each evaluating the overall quality, including (1) editing completeness and accuracy, (2) original image preservation, (3) 3D consistency, and (4) image appearance, and return the scores of all baselines as a JSON array. We repeat multiple times and take the average.
    - Our GPT score can be regarded as a Monte-Carlo implementation of the recently proposed "VQAScore" [*Evaluating Text-to-Visual Generation with Image-to-Text Generation*, In ECCV'24], a metric based on a vision-language model's evaluation of the generated image, which has been shown to outperform CLIP-based scores.
    - As the vision-language model is the only model that is powerful enough to understand and evaluate all the aspects, this VQA-based metric, along with our GPT score, can be viewed as a relatively complete automated quantitative measurement compared to CLIP-based scores.

---

### Decision · Program_Chairs · 2024-09-25

**Decision:**

Accept (poster)

**Comment:**

This paper aims at 3D scene editing given an instruction prompt. The main idea is to decompose the editing task into subtasks with similar difficulty. With such decomposition, the editing task becomes more manageable and introduces less inconsistency. During rebuttal, reviewers raised concerns about quantitative evaluation, missing ablation studies and questions about methodology. Most concerns are addressed during rebuttal and all reviewers lean to accept the paper. Thus the paper is recommended for acceptance, we encourage the authors to include additional evaluations, ablation studies in the revised paper.